# Guided Internet-Based Cognitive Behavioral Therapy for Insomnia: Prognostic and Treatment-Predictive Factors

**DOI:** 10.3390/diagnostics13040781

**Published:** 2023-02-19

**Authors:** Polina Pchelina, Simone B. Duss, Corrado Bernasconi, Thomas Berger, Tobias Krieger, Claudio L. A. Bassetti, Antoine Urech

**Affiliations:** 1Department of Neurology and Neurosurgery, I.M. Sechenov First Moscow State Medical University, 119991 Moscow, Russia; 2Interdisciplinary Sleep-Wake-Epilepsy-Center and Swiss Sleep House Bern, Inselspital, Bern University Hospital, University of Bern, 3010 Bern, Switzerland; 3Department of Neurology, Inselspital, Bern University Hospital, University of Bern, 3010 Bern, Switzerland; 4Departement of Clinical Psychology and Psychotherapy, Institute of Psychology, University of Bern, 3012 Bern, Switzerland

**Keywords:** chronic insomnia, internet-based intervention, cognitive behavioral therapy of insomnia, sleep restriction

## Abstract

Understanding which factors predict the outcome of internet-based cognitive behavioral therapy for insomnia (iCBT-I) may help to tailor this intervention to the patient’s needs. We have conducted a secondary analysis of a randomized, controlled trial comparing a multicomponent iCBT-I (MCT) and an online sleep restriction therapy (SRT) for 83 chronic insomnia patients. The difference in the Insomnia Severity Index from pre- to post-treatment and from pre-treatment to follow-up at 6 months after treatment was the dependent variable. Prognostic and treatment-predictive factors assessed at baseline were analyzed with multiple linear regression. The shorter duration of insomnia, female gender, high health-related quality of life, and the higher total number of clicks had prognostic value for a better outcome. Other factors were found to be prognostic for outcome at the follow-up assessment: treatment with benzodiazepines, sleep quality, and personal significance of sleep problems. A high level of dysfunctional beliefs and attitudes about sleep (DBAS) was a moderator for better effects in the MCT at post-treatment assessment. Various prognostic factors (e.g., duration of insomnia, gender, or quality of life) may influence the success of treatment. The DBAS scale may be recommended to select patients for MCT rather than SRT.

## 1. Introduction

Cognitive behavioral therapy for insomnia (CBT-I) is the gold standard for treating chronic insomnia [1,2]. The treatment targets maladaptive behaviors and dysfunctional cognition that perpetuate sleep problems and is one of the most effective treatments for insomnia [3]. The multicomponent CBT-I typically includes the interventions of stimulus control, sleep restriction, sleep hygiene, relaxation techniques, and cognitive restructuring [4,5]. Nevertheless, access to it is often limited by the lack of trained therapists and the cost of the treatment [6,7]. The majority of physicians usually prescribe pharmacological treatment [8,9]. A large number of meta-analyses and reviews have shown that chronic insomnia can be successfully managed with internet-based CBT-I (iCBT-I) programs with or without therapist guidance [10,11]. iCBT-I programs solve the problem of a shortage of CBT-I specialists and allow researchers to acquire data from larger samples, improving the generalizability of results [12,13]. However, online approaches do not produce significant improvement in 30–40% of chronic insomnia patients [14]. 

There appears to be a phenotypic heterogeneity of insomnia disorder that is not related to sleep quality and quantity alone (e.g., emotional dysregulation, pre-sleep arousal, or patient characteristics) [15]. Therefore, more knowledge is needed on which factors and under which conditions might improve the treatment of insomnia. Objective sleep characteristics did not consistently predict treatment success [16]. The effect of demographic variables such as sex, age, education, and outcome expectancy, psychiatric comorbidities on treatment outcome is inconsistent. There is some evidence of a negative effect of younger age on improvement while some studies showed no effect of age [17,18]. The same discordance exists regarding the effects of comorbid major depression and anxiety [18,19,20,21,22]. The most stable predictive effects are observed for cognitive, emotional, and behavioral factors related to sleep: sleep-threat monitoring, dysfunctional beliefs, safety behaviors, sleep-related worry, and pre-sleep arousal [17,18,23,24].

Although multicomponent intervention shows higher remission rates and is recommended by the leading sleep medicine organizations [25], some studies report the effects of specific components of CBT-I. For example, behavioral interventions produce faster but less enduring effects, while cognitive interventions are delayed in action but lead to more sustained outcomes [26]. On the single techniques level, there is evidence that sleep restriction is one of the most effective CBT-I components, whereas psychoeducation and sleep hygiene, as stand-alone elements, are unlikely to be effective as individual therapy [27,28,29]. However, data are too limited to draw firm conclusions about predictors of the effect of the single components of CBT-I and iCBT-I. With increasing options for treating insomnia, there is a need to predict which patients are more likely to benefit from a more complex multicomponent treatment and who profits from a simplified intervention. This knowledge would help to create algorithms tailoring iCBT-I programs to the needs and individual sleep and non-sleep features of a particular patient. This exploratory analysis focuses on patients’ characteristics that may predict the course of insomnia during treatment and predict the effect of either multicomponent iCBT-I (MCT) or internet-based sleep restriction therapy (SRT). These factors include demographics, personality characteristics, and mental, physical, and sleep health history. 

## 2. Materials and Methods

### 2.1. Study Design and Participants

Data for the analysis were collected during a primary three-arm randomized controlled trial comparing two intervention groups (internet-based multi-component cognitive behavioral self-help intervention and internet-based sleep restriction intervention for insomnia) to a usual care group as a control group. The detailed protocol of the study recruitment, intervention description, and outcomes have been previously reported elsewhere [30]. The trial was registered at www.clinicaltrials.gov (NCT03110263) and was approved by the Ethics Committee of the Canton of Bern, Switzerland (2016-00295). A total of 104 participants with chronic insomnia according to the International Classification of Sleep Disorders criteria [31] were recruited in 2016–2017, including 83 participants in active treatment groups. Informed consent was obtained from all subjects involved in the primary study.

The interventions were 8-week guided iCBT-I programs. MCT consisted of eight text-based sessions and tasks and was based on psychoeducational, behavioral (sleep restriction, stimulus control, and relaxation), and cognitive (belief restructuring) interventions. SRT consisted of sleep restriction instructions that are embedded in an introductory and psychoeducational module. Guidance was provided on a weekly basis in both intervention arms via E-mail. All participants had access to any other healthcare resources during their study participation.

### 2.2. Dependent Variable

The primary outcome of the present analysis was the Insomnia Severity Index (ISI), with improvement calculated as change from pre- to post-treatment, i.e., higher values of this change variable were associated with better effects. The ISI is a seven-item insomnia assessment tool examining both nighttime and daytime aspects of insomnia disorder and is sensitive to change. The 5-point Likert scale is used to rate each item (e.g., 0 = no problem; 4 = very severe problem), yielding a total score ranging from 0 to 28. The German version of the ISI has shown acceptable psychometric properties [32]. Higher scores on the ISI indicate more severe insomnia. 

### 2.3. Prognostic and Treatment-Predictive Factors

A prognostic factor describes a measure that is related to the clinical outcome when therapy is applied. It can also be considered a measure of disease progression. In contrast, a predictive factor is a measure that is associated with the response or non-response of a therapy. Moreover, prognostic factors of improvement are not necessarily predictors of treatment effect [33]. 

#### 2.3.1. Baseline Demographic and Medical History Variables

The baseline characteristics of the study participants in the whole sample are shown in Table 1. Demographic and medical history information was collected at baseline. Age, level of education, and body mass index (BMI) were treated as continuous variables. Level of education was presented as a four-level scale from 1 «no education» to 4 «university education». Consumption of medical help within the last 3 months and treatment with benzodiazepines, antidepressants, and natural medicines were coded as separate categorical variables with two levels (yes or no). Sex was coded as 1 = male and 2 = female. The duration of insomnia was coded as 1 = 3–12 months; 2 = More than 12 months. Marital status was coded as a categorical variable with four levels: single, having a partner, divorced, and married. Employment was coded as a categorical variable with five levels: retired, homemaker, student, full-time employed, and part-time employed.

#### 2.3.2. Variables on Sleep and Psychopathology

We included baseline scores of the sleep, psychopathology, and well-being questionnaires in our analysis to evaluate their prognostic and predictive value. Overall sleep quality was evaluated with the Pittsburgh Sleep Quality Index (PSQI) [34,35]. Maladaptive beliefs in insomnia were assessed in the 16-item version of the Dysfunctional Beliefs and Attitudes about Sleep (DBAS) scale [36,37]. To measure the depressive symptoms, the German short version of the Center for Epidemiological Studies Depression Scale [38], the “Allgemeine Depressions-Skala—Kurzform” (ADS-K) was used [39]. Participants reported their quality of life in the quality of life questionnaire (QoL). They also rated their overall health on a visual analog scale (QoL-VAS) from 0 (the worst health you can imagine) to 100 (the best health you can imagine) [40]. The personal significance of the individual’s sleep problems was assessed in a generated questionnaire with a general score varying from 10 to 70. This questionnaire comprises 10 questions about characterizing the importance of sleep problems and recovery, self-confidence, and autonomy in the handling of sleep problems. Participants rated their expectations about treatment success (At this point, how successful do you think the online therapy will be in relieving your sleep disorder?; scale from 1 “absolutely unsuccessful” to 9 “very successful”). For the evaluation of subjective sleep-related measures, we used sleep diary data as sleep onset latency (SOL); total sleep time (TST); and sleep efficiency (SE), calculated as TST/time in bed × 100; and wake after sleep onset (WASO) was derived from the sleep diary that was collected for the first week, and averaged.

#### 2.3.3. Intervention Adherence 

Treatment adherence was included in the analysis since it has been shown to be a predictor of the internet-delivered therapy effect [41]. For this purpose, we selected two continuous variables reflecting the patient’s activity in the program. Time spent online shows how much time participants spent logged in to the program. The total number of clicks is the sum of all clicks participants made in both arms. 

### 2.4. Statistical Methods

The analysis was performed using the R statistical software. Baseline characteristics were compared between the two groups by t-test for independent samples. The statistical techniques used for the main analysis of data in this study are two-way ANOVA and multiple linear regression. A set of baseline variables were assessed for their prognostic and predictive value for the ISI improvement through multiple linear regression models for the whole sample. Two-way ANOVA was conducted for categorical variables with more than 2 levels to see if there is overall interaction. Variables ISI post-treatment and follow-up, DBAS, ADS, QoL, QoL-VAS, personal significance of sleep problems, success expectancy, SE, TST and WASO were checked for missing data. Missing values were handled with the use of R package mice for multiple imputations with the number of multiple imputations equal to 5 and the number of iterations equal to 50 [42]. Since the rate of missing data was 15% for ISI improvement post-treatment and 28% for ISI improvement at follow-up, multiple imputations were used for all the statistical models. The prognostic analysis assesses characteristics associated with the overall outcome over time regardless of intervention. In a prognostic analysis of ISI improvement, various baseline variables were assessed in separate models adjusted for ISI at baseline and treatment conditions (MCT or SRT). In a predictive analysis, baseline factors associated with the treatment effect were investigated by a “condition by predictor” interaction term added to the models. Since the ISI score at baseline explained a large fraction of the variance of the final score, it was included in all models. Models where the p-value for the predictor variable:condition interaction effect was <0.1 were further assessed (e.g., graphically). Otherwise, the significance level of statistical tests was set to 5%, and no correction for multiplicity was applied.

## 3. Results

### 3.1. Baseline Differences

MCT and SRT participants did not differ in baseline demographic characteristics or other variables investigated as potential predictive and prognostic factors (see Table 1). There were significant differences between MCT and SRT groups on DBAS scores (*p* = 0.04), sleep-diary-derived sleep latency (*p* = 0.03), the total number of clicks (*p* = 0.04), and time spent online (*p* = 0.01). 

### 3.2. Prognostic Analysis

#### 3.2.1. ISI Change from Pre-Treatment to Post-treatment

The following prognostic variables showed no significant association with change in ISI score between pre- and post-treatment: age, BMI, marital status, employment, education, consumption of medical care within the last 3 months, treatment with benzodiazepines, treatment with antidepressants, treatment with natural medicines, PSQI, DBAS, ADS, QoL, success expectancy, personal significance of sleep problems, time spent online, SE, TST, SOL, and WASO (Table 2). We observed a significant relationship for the female gender, which provided 2.20 additional score points to the ISI compared to males (*p* = 0.032). The severity of insomnia decreased more prominently in females regardless of which group they were randomized in (see Figure 1). A shorter duration of insomnia provided 3.27 additional score points to the ISI change as compared to insomnia longer than 1 year (*p* = 0.011) (see Figure 1). Participants, who evaluated their health-related quality of life higher on the QoL-VAS at baseline, improved more. Every additional 10 points on this score led to a 0.64 change of the ISI (*p* = 0.036) independent of the group allocation (see Figure 1). A higher number of clicks within 8 weeks was a significant prognostic factor for improvement. Every additional 100 clicks in the online program increased the ISI difference by 0.178 score points (*p* = 0.029) (see Figure 1).

#### 3.2.2. ISI Change from Pre-Treatment to Follow-Up

Regarding age, sex, BMI, marital status, employment, education, consumption of medical care within the last 3 months, treatment with antidepressants, treatment with natural medicines, duration of insomnia, DBAS, ADS, QoL, QoL-VAS, success expectancy, number of clicks, time spent online, SE, TST, SOL, and WASO, we observed no significant associations with the change in insomnia symptoms from pre- to follow-up treatment. Treatment with benzodiazepines at baseline assessment was associated with worse ISI dynamic at follow-up. Participants who used benzodiazepines within 3 months before the study improved by 2.51 score points less than the rest (*p* = 0.014) (see Figure 2). A higher baseline PSQI score (characterizing the worse quality of sleep) was associated with worse outcomes after 6 months of follow-up. With each score point increase in the level of PSQI ISI change decreased by 0.365 (*p* = 0.0073) (see Figure 2). Every additional score point of the personal significance of the sleep problems scale increased the ISI change by 0.08 score points (*p* = 0.044) (see Figure 2).

### 3.3. Predictive Analysis

#### ISI Change from Pre-Treatment to Post-Treatment and from Pre-Treatment to Follow-Up

Among such parameters as age, sex, marital status, employment, education, consumption of medical care within the last 3 months, treatment with benzodiazepines, treatment with antidepressants, treatment with natural medicines, duration of insomnia, PSQI, ADS, QoL-VAS, Success expectancy, number of clicks, time spent online, SE, TST, SOL, and WASO, we observed no significant predictors. The overall lower quality of life score at baseline was a nearly significant predictor of worse ISI change from pre-treatment to follow-up. Every score point of quality of life was associated with 3.293 score points less ISI change (*p* = 0.092). Participants having stronger dysfunctional beliefs about sleep benefited more from MCT than from sleep restriction. Every additional score point on the DBAS at baseline provided an additional decrease in ISI by 0.097 score points in the MCT group (*p* = 0.04). In contrast, those who had a lower level of beliefs benefited more from sleep restriction than from MCT (see Figure 3). However, the baseline DBAS score of the two groups was different: the DBAS average score in the SRT group was 70.64, in MCT 80.63 (t = −2.12; *p* = 0.03). The exploratory character of our analysis enables us to mention nearly significant predictors of effect, namely weight and BMI. Participants with higher BMI demonstrated small to no effect in the SRT group while in the MCT group every 1 kg/m^2^ was associated with a 0.39 (*p* = 0.09.) decrease in ISI score accordingly. The tendency of BMI effect remained after 6 months of follow-up, when every 1 kg/m^2^ was associated with a 0.311 (*p* = 0.08.) decrease in ISI score (Table 3).

## 4. Discussion

The choice of therapy for chronic insomnia in clinical practice usually requires time and effort. In case the patient fails to remit with first-stage therapy, the clinician switches from one treatment approach to another on a trial-and-error basis. Every treatment failure postpones remission. Therefore, it is critical to be able to predict an individual patient’s ability to improve sleep via a given treatment. Furthermore, it is advantageous to be able to select the appropriate treatment option for an individual patient based on established predictive factors. Our study aimed to assess characteristics with prognostic value for the improvement regardless of the treatment approach and characteristics associated with better response to either sleep restriction therapy or multicomponent treatment modality. Additionally, we confirmed a positive association between treatment effect and adherence. One of the strengths of the present study was a sample with unrestricted access to healthcare resources, which makes the results more generalizable.

Our study has indicated that the female gender variable was a significant prognostic factor of the ISI improvement after 8 weeks regardless of treatment modality. Previous studies found that sex was never featured as a significant predictive factor of the insomnia therapy effect [16]. Moreover, observational studies of the natural course of insomnia demonstrated either no effect of sex or a higher insomnia persistence rate over 3 years in women [43,44]. As we know from the previous studies, longer insomnia duration was associated with a poorer outcome of CBT-I interventions [45]. This finding was replicated in our study, which can be explained by the fact that long duration of sleep problems is associated with higher severity of insomnia in general. Higher health-related quality of life appeared to be a prognostic factor of sleep improvement. These results are inconsistent with those of several previous studies that found a positive effect of low physical health-related quality of life for pre- to post-treatment and pre- to post-follow-up improvements in insomnia severity, i.e. lower physical well-being at baseline predicted larger improvements on sleep [20]. The high personal significance of sleep problems and worse sleep quality were oppositely directed prognostic factors of change in insomnia symptoms. Worse subjective perception of sleep evaluated in various questionnaires is expected to increase a patient’s motivation to adhere to treatment and is a predictor of better treatment outcomes as shown in studies on CBT-I [20,21,22]. However, our study showed the opposite effect, i.e., worse subjective characteristics of sleep were prognostic for low ISI difference. This result can be explained by the fact that in addition to insomnia symptoms, this questionnaire evaluates several dimensions of sleep including sleep breathing disorders, parasomnias, and the use of sleep aids. Moreover, our analysis yielded that personal significance was a significant prognostic factor of ISI change. Other studies revealed that the high personal significance of sleep problems can reflect insomnia severity which was established as a predictor of effect in other CBT-I studies [21,46,47]. Furthermore, this variable depends on individual beliefs (e.g., beliefs about the ability to deal with my sleep disorder) and locus of control. These cognitive factors were shown to be mediators of change of the subjective evaluation of insomnia severity (Insomnia Severity Index) and sleep diary outcomes in another study of iCBT-I [22]. Adherence to the treatment is an important mediator of effect in cognitive behavioral therapy in general and in technology-mediated treatments particularly [48]. The internet-based approach can measure logged adherence–objective data about the activity of participants on the website represented in two variables: time spent online and the number of clicks. In our study, the total number of clicks was a significant prognostic factor of the treatment effect. This result confirms that patients should be encouraged not only to log in but also to remain active on the iCBT-I website or app by guiding specialists or automatically. 

A series of studies have shown the added value of CBT-I in combination with hypnotics or antidepressants although its long-term effect seems to be unstable [14,49]. In our study participants receiving treatment with benzodiazepines within 3 months before the study improved less than the rest. This finding is in line with the previous comparisons of the effectiveness of CBT-I vs the combination of zolpidem and CBT-I [14,50]. It showed that the addition of medication to CBT-I added modest benefits in terms of increased sleep time after the CBT-I course, but a better long-term outcome was achieved when hypnotic was discontinued during or directly after the CBT-I course. Guidelines suggest that medication discontinuation allows patients to integrate newly learned psychological and behavioral skills to overcome insomnia, while benzodiazepines may sometime mask the effect of CBT-I [51]. In our analysis, pharmacotherapy in general and treatment with antidepressants or natural medicines were not significant prognostic factors, which could be explained by their modest effect on sleep and a broader spectrum of action by targeting mood symptoms. 

Our study revealed that prognostic factors found to be significant after treatment differ from those after 6 months of follow-up. The inconsistency of some prognostic factors can be explained by their nature. The fact that the effect of benzodiazepine use before treatment was not registered post-treatment but appeared after a 6-month interval could be partly explained by the fact that these medicaments are usually prescribed for a short period. Thus, by the post-treatment measurement, the benzodiazepine course could still be in progress providing additional benefits for sleep. By the follow up most patients could already establish tolerance to benzodiazepines or discontinue treatment with benzodiazepines and experience withdrawal symptoms or rebound of insomnia [52]. The tendency that insomnia may progress with time could partly explain the instability of the prognostic effect for the duration of insomnia. 

Finding predictors of the effect of different modalities of iCBT-I is a comparatively new approach. A previous study showed that dysfunctional beliefs about sleep, locus of control, sleep-related worry, and pre-sleep arousal have the most reliable evidence for a role as a mediating factor of online CBT-I effect in comparison with wait-list [52,53] or online patient education condition [47]. Our analysis revealed that baseline dysfunctional beliefs and attitudes about sleep score were the only significant factor predicting the effect of different modalities of iCBT-I. Patients with a higher level of misconceptions about sleep benefited from the MCT approach that included cognitive restructuring interventions (e.g., targeting unrealistic beliefs about sleep). This result supplements the results of the initial analysis where the MCT group benefited significantly more than the SRT group regarding dysfunctional sleep-related beliefs [30]. The participants who had a low level of dysfunctional beliefs and attitudes about sleep benefited more from the approach that let them concentrate on the sleep regimen—SRT. Profiling patients by their levels of dysfunctional beliefs and attitudes about sleep may be useful within the stepped care system for insomnia. The use of DBAS in primary care will help to distinguish patients with low and high level of dysfunctional beliefs. While those with a low DBAS score are expected to benefit from more straightforward behavioral recommendations (e.g., sleep restriction), which are easy to deliver via the internet, the patients having misconceptions of sleep need and role in everyday life will require multicomponent approach and closer attention of a specialist, including face-to-face consultations. 

Some important limitations of the present study have to be considered. First, the limitations of the design of the initial study made us unable to test the prognostic and predictive effect of baseline variables in three groups including the control group because of the small sample size of this group. Second, post hoc analyses are as a rule connected with the problem of multiple comparisons, and thus we emphasize that the present findings are strictly exploratory and should be interpreted with caution. Third, since the participants were recruited from the community and was self-selected and it is plausible those predictors of outcome could be different with patients who seek medical help actively.

## 5. Conclusions

Although the standard multicomponent CBT-I is considered to be the treatment of choice, it can and must be tailored to the needs of the individual patient [26]. Identification of key predictors of treatment outcomes such as dysfunctional beliefs and attitudes about sleep score may help to guide decision makers and clinical practitioners regarding whether non-pharmacological treatment of insomnia should be primarily behavioral, such as sleep-restriction therapy or multicomponent therapy or whether the patient should be referred to a specialized healthcare facility. Future studies should include larger samples and use a stratified sampling approach to test whether different factors such as tailored treatment programs, guidance, support, or interactive activities improve outcomes and treatment. 

In sum, the present analysis enhances the knowledge of the applicability of iCBT-I and its components. The findings have potential implications for a broader variety of internet interventions.

## Figures and Tables

**Figure 1 diagnostics-13-00781-f001:**
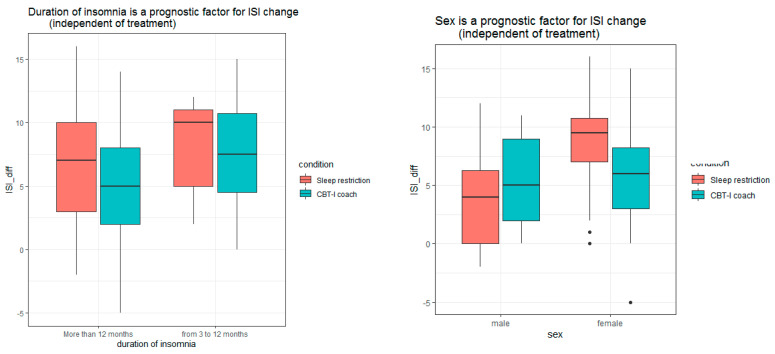
Prognostic value of duration of insomnia, sex, health-related quality of life, total number of clicks for ISI change after treatment course. ISI—insomnia severity index, CBT-I coach—internet-delivered multicomponent cognitive behavioral therapy for insomnia, ISI_diff—the difference of the ISI from pre-to post-treatment (ISI pre-treatment–ISI post-treatment), QoL VAS—health-related quality of life measured with visual analog scale from 0 to 100.

**Figure 2 diagnostics-13-00781-f002:**
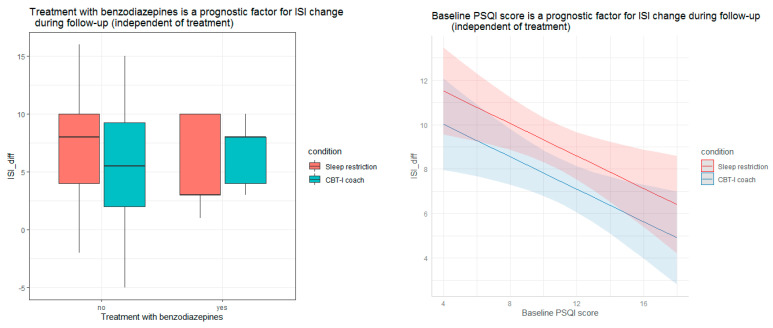
Prognostic value of treatment with benzodiazepines, baseline PSQI score, and baseline personal significance of sleep problems for ISI change after follow up. ISI—insomnia severity index, CBT-I coach—internet-delivered multicomponent cognitive behavioral therapy for insomnia, ISI_diff—the difference of the ISI from pre-to post-treatment (ISI pre-treatment–SI post-treatment), PSQI—Pittsburgh Sleep Quality Index, ES—personal significance of sleep problems.

**Figure 3 diagnostics-13-00781-f003:**
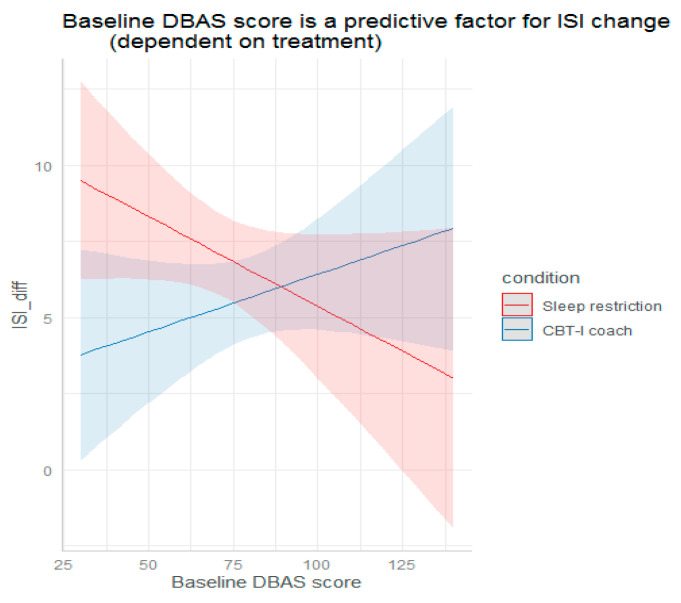
Predictive value of baseline dysfunctional beliefs and attitudes about sleep for ISI change in different arms. ISI—insomnia severity index, CBT-I coach—internet-delivered multicomponent cognitive behavioral therapy for insomnia, ISI_diff—the difference of the ISI from pre-to post-treatment (ISI pre-treatment–ISI post-treatment), DBAS—dysfunctional beliefs and attitudes about sleep scale.

**Table 1 diagnostics-13-00781-t001:** Baseline characteristics used in analysis of prognostic and predictive factors of ISI change of 83 study participants with CI.

	MCT *n* = 42	SRT *n* = 41	Statistic
Age, years; mean, SD	42.17 (12.40)	46.59 (17.52)	t = −1.32, *p* = 0.19
Body mass index, kg/m^2^; mean, SD	23.60 (3.64)	24.23 (5.20)	t = −0.64, *p* = 0.52
Sex, *n* (%)			
Female	26 (61.9%)	28 (68.3%)	X^2^ = 0.14, *p* = 0.70
Male	16 (38.1%)	13 (31.7%)	
Consumption of medical help within the last 3 months, *n* (%)			
yes	16 (38.1%)	13 (31.7%)	X^2^ = 0.14, *p* = 0.70
no	26 (61.9%)	28 (68.3%)	
Treatment with benzodiazepines, *n* (%)			
yes	9 (22)	5 (11.9)	X^2^ = 0.86, *p* = 0.35
no	32 (78)	37 (88.1)	
Treatment with antidepressants, *n* (%)			
yes	3 (7.3)	4 (9.5)	X^2^ < 0.01, *p* = 1.00
no	38 (92.3)	38 (90.5)	
Treatment with natural medicines, *n* (%)			
yes	5 (12.2)	11 (26.2)	X^2^ = 1.79, *p* = 0.18
no	36 (87.8)	31 (73.8)	
Duration of insomnia, *n* (%)			
More than 3 months	5 (11.9%)	10 (24.4%)	X^2^ = 1.42, *p* = 0.23
More than 1 year	37 (88.1%)	31 (75.6%)	
Marital status, *n* (%)			
Single	11 (26.2%)	9 (22.0%)	X^2^ = 3.26, *p* = 0.35
Having a partner	9 (21.4%)	13 (31.7%)	
Divorced	2 (4.8%)	5 (12.2%)	
Married	20 (47.6%)	14 (34.1%)	
Widowed	0 (0.0%)	0 (0.0%)	
Employment, *n* (%)			
Retired	1 (2.4%)	4 (9.8%)	X^2^ = 4.11, *p* = 0.39
Homemaker	0 (0.00%)	0 (0.00%)	
Student	5 (11.9%)	6 (14.6%)	
Full-time employed	24 (57.1%)	17 (41.5%)	
Part-time employed	12 (28.6%)	13 (31.7%)	
Education, *n* (%)			
No education	0 (0.0%)	2 (4.9%)	X^2^ = 2.35, *p* = 0.50
Vocational training	11 (26.2%)	12 (29.3%)	
Higher secondary school	4 (9.5%)	3 (7.3%)	
University education	27 (64.3%)	24 (58.5%)	
ISI; mean, SD	16.20 (3.75)	17.37 (3.44)	t = −1.47, *p* = 0.15
PSQI; mean, SD	10.12 (3.13)	11.05 (3.14)	t = −1.35, *p* = 0.18
DBAS; mean, SD	70.63 (21.83)	80.63 (21.40)	t = −2.12, *p* = 0.04
ADS-K; mean, SD	12.83 (6.33)	13.05 (5.42)	t = −0.04, *p* = 0.97
QoL_total; mean, SD	74.27 (17.62)	73.00 (15.25)	t = −0.45, *p* = 0.65
QoL-VAS; mean, SD	1.37 (0.40)	1.4 (0.37)	t = 0.32, *p* = 0.75
Success expectancy; mean, SD	5.07 (1.88)	5.29 (1.59)	t = −0.65, *p* = 0.52
Personal significance of sleep problems; mean, SD	51.9 (10.45)	52.32 (9.48)	t = −0.05, *p* = 0.96
Sleep efficiency, %; mean, SD	76.02 (13.89)	77.49 (11.35)	t = 0.53, *p* = 0.60
Total sleep time, minutes; mean, SD	364.05 (72.61)	371.47 (57.75)	t = 0.51, *p* = 0.61
Sleep latency, minutes; mean, SD	40.83 (43.21)	24.32 (23.39)	t = −2.16, *p* = 0.03
Wake after sleep onset, minutes; mean, SD	50.28 (40.92)	55.53 (46.09)	t = 0.55, *p* = 0.58
Time spent online, minutes; median (Q1;Q3)	481 (282; 907.35)	392 (183.43; 591.57)	t = 2.58, *p* = 0.01
Total number of clicks; mean, SD	1470.76 (1535.67)	1064.0 (664.06)	t = 2.124, *p* = 0.04

ISI—insomnia severity index, CI—chronic insomnia, SD—standard deviation, Q1—25th percentile, Q3—75th percentile, PSQI—Pittsburgh Sleep Quality Index, DBAS—Dysfunctional Beliefs about Sleep scale, ADS-K—Allgemeine Depressions-Skal—Kurzform, QoL—total, quality of life, QoL-VAS—quality of life, health facet on visual analog scale, t—Student’s t test statistic, X^2^—chi square test statistic facet on visual analog scale.

**Table 2 diagnostics-13-00781-t002:** Prognostic regression analysis coefficients. A *p*-value indicates a significance of relationship between the baseline variable and ISI improvement post-treatment. The greater ISI improvement reflecting improvement after treatment is characterized by the higher estimate. In the prognostic analysis columns estimates, standard errors, and *p*-values are presented for the main effect of the analyzed variable.

	Prognostic Analysis ISI_diff1~ISI_t0 + Var_Predictor + Condition		Prognostic Analysis ISI_diff2~ISI_t0 + Var_Predictor + Condition	
	Estimate (SE) for the Main Effect of Variable	*p* Value	Estimate (SE) for the Effect of Variable	*p* Value
Intercept	0.79 (2.32)	0.735	−0.60 (1.82)	0.743 *
ISI_total_t0	0.33 (0.14)	0.018 *	0.53 (0.11)	3.82 × 10^−6^ *
Condition	−1.30 (0.98)	0.189	−1.65 (0.75)	0.039 *
Age, years	0.01 (0.03)	0.855	0.01 (0.76)	0.814
Sex, male (29)	1.00 (0.00)	0.032 *	1.00 (0.00)	0.153
female (54)	2.20 (1.00)		1.13 (0.78)	
Weight	−0.06 (0.03)	0.069	−0.01 (0.03)	0.653
BMI	−0.15 (0.11)	0.180	−0.06 (0.08)	0.516
Marital status	F = 1.763	0.161	F = 1.45	0.080
Employment	F = 1.614	0.179	F = 0.97	0.428
Education	0.65 (0.53)	0.225	0.02 (0.41)	0.964
Consumption of medicine last 3 months				
yes (29)	1.00 (0.00)	0.960	1.00 (0.00)	0.174
no (54)	−0.05 (1.03)		1.07 (0.78)	
Treatment with benzodiazepines				
yes (14)	1.00 (0.00)	0.368	1.00 (0.00)	0.014 *
no (69)	−1.23 (1.35)		−2.51 (1.01)	
Treatment with antidepressants				
yes (7)	1.00 (0.00)	0.350	1.00 (0.00)	0.096
no (76)	1.52 (1.75)		2.23 (1.32)	
Treatment with natural medicine				
yes (16)	1.00 (0.00)	0.790	1.00 (0.00)	0.797
no (67)	0.33 (1.26)		0.25 (0.96)	
Duration of insomnia				
3 to 12 months (15)	1.00 (0.00)	0.011 *	1.00 (0.00)	0.293
More than 1 year (68)	−3.27 (1.27)		1.07 (1.07)	
PSQI	−0.12 (0.18)	0.490	−0.37 (0.13)	0.007 *
DBAS	−0.01 (0.03)	0.830	0.002 (0.02)	0.277
ADS	0.105 (0.09)	0.229	−0.10 (0.07)	0.156
QoL	−1.33 (1.36)	0.330	−0.14 (1.03)	0.167
QoL health related VAS, 0–100	0.06 (0.03)	0.036 *	0.03 (0.02)	0.162
Success expectancy, 0–9	0.37 (0.29)	0.200	0.21 (0.22)	0.339
Personal significance of sleep problems, 10–70	0.01 (0.05)	0.913	0.08 (0.04)	0.044 *
Total number of clicks	0.002 (0.001)	0.029 *	0.0005 (0.001)	0.477
Time spent online, hours	0.09 (0.05)	0.092	0.07 (0.04)	0.081
Sleep efficiency, %	0.001 (0.004)	0.988	0.02 (0.03)	0.513
Total sleep time, min	0.01 (0.01)	0.452	−0.003 (0.01)	0.632
Sleep onset latency, min	0.002 (0.02)	0.910	−0.01 (0.01)	0.623
Wake after sleep onset, min	−0.01 (0.01)	0.353	−0.0004 (0.01)	0.963

Note: * *p* < 0.05.

**Table 3 diagnostics-13-00781-t003:** Predictive regression analysis coefficients. A *p*-value indicates a significance of relationship between the baseline variable and ISI improvement post-treatment. The greater ISI improvement reflecting improvement after treatment is characterized by the higher estimate. In the prognostic analysis columns estimates, standard errors, and *p*-values are presented for the effect of interaction predictor * condition.

	Predictive Analysis ISI_diff1~ISI_t0 + Var_Predictor * Condition		Predictive Analysis ISI_diff2~ISI_t0 + Var_Predictor * Condition	
	Estimate (SE) for the Effect of Interaction Variable: Condition	*p* Value	Estimate (SE) for the Effect of Interaction Variable: Condition	*p* Value
Intercept	0.79 (2.32)	0.735	−0.60 (1.82)	0.743 *
ISI_total_t0	0.33 (0.14)	0.018 *	0.53 (0.11)	3.82 × 10^−6^ *
Condition	−1.30 (0.98)	0.189	−1.65 (0.75)	0.039 *
Age, years	0.01 (0.07)	0.873	0.03 (0.05)	0.590
Sex, male (29)	1.00 (0.00)	0.102	1.00 (0.00)	0.257
female (54)	−3.30 (1.99)		−1.79 (1.57)	
Weight	0.11 (0.07)	0.090	0.09 (0.05)	0.081
BMI	0.39 (0.23)	0.089	0.31 (0.18)	0.080
Social status	F = 1.152	0.334	F = 1.71	0.172
Working status	F = 0.576	0.632	F = 0.465	0.708
Education	−1.54 (1.05)	0.149	−0.40 (0.82)	0.632
Medications use				
yes (29)	1.00 (0.00)	0.96	1.00 (0.00)	0.745
no (54)	−2.51 (2.04)		0.51 (1.56)	
Benzodiazepines				
yes (14)	1.00 (0.00)	0.249	1.00 (0.00)	0.834
no (69)	3.13 (2.69)		−0.42 (2.02)	
Antidepressants				
yes (7)	1.00 (0.00)	0.109	1.00 (0.00)	0.401
no (76)	5.64 (3.48)		2.26 (2.67)	
Natural medicine				
yes (16)	1.00 (0.00)	0.160	1.00 (0.00)	0.156
no (67)	3.69 (2.61)		2.86 (2.00)	
Duration of insomnia				
3 to 12 months (15)	1.00 (0.00)	0.704	1.00 (0.00)	0.982
More than 1 year (68)	0.98 (2.57)		0.05 (2.04)	
PSQI	−0.05 (0.32)	0.878	−0.001 (0.23)	0.998
DBAS	0.10 (0.05)	0.040 *	0.04 (0.04)	0.345
ADS	−0.26 (0.17)	0.12	−0.14 (0.13)	0.270
QoL	−3.73 (2.54)	0.146	−3.29 (1.93)	0.092
QoL health-related VAS, 0–100	−0.02 (0.06)	0.740	0.03 (0.05)	0.520
Success expectancy, 0–9	−0.33 (0.57)	0.564	0.10 (0.44)	0.820
Personal significance of sleep problems, 10–70	0.002 (0.10)	0.986	0.07 (0.07)	0.323
Total number of clicks	−0.0005 (0.002)	0.795	−0.001 (0.001)	0.546
Time spent online, hours	0.08 (0.18)	0.655	0.05 (0.13)	0.741
Sleep efficiency, %	0.05 (0.08)	0.566	0.02 (0.06)	0.731
Total sleep time, min	0.01 (0.02)	0.550	−0.01 (0.01)	0.539
Sleep onset latency, min	−0.01 (0.03)	0.810	−0.01 (0.03)	0.746
Wake after sleep onset, min	−0.03 (0.02)	0.233	−0.03 (0.02)	0.132

Note: * *p* < 0.05.

## Data Availability

No new data were created or analyzed in this study. Data sharing is not applicable to this article.

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
