# Peer review of "Guided Internet-Based Cognitive Behavioral Therapy for Insomnia: Prognostic and Treatment-Predictive Factors"

_diagnostics, 2023, doi:10.3390/diagnostics13040781_

Round 1
Reviewer 1 Report
An overall good job that needs to be improved with only few changes regarding English languageAuthor Response
Response to Reviewer 1 Comments
Point 1: An overall good job that needs to be improved with only few changes regarding English language
Response 1: Thank you very much for your positive assessment. The manuscript was carefully revised to improve the quality of english spelling. We hope that you are satisfied with our response. Thanks again for taking your time.
Reviewer 2 Report
Dear Authors,
Generally, this study was interesting and well-conducted. I would recommend a minor revision to this manuscript. I summarize my comments below. Thank you for allowing me to review this manuscript for you.
1. Please use an unstructured form for the abstract per journal requirements. Please remove unnecessary abbreviations if they were not used more than three times. Line 29: Did you mean an "MCT" rather than an SRT?
2. The introduction was extraordinarily long. Please make it more concise. May consider providing a table that summarizes the difference between iCBT-I and CBT-I, and MCT and SRT. Otherwise, please use the same abbreviations throughout the manuscript.
3. Please provide detailed information on the statistical software (R software?).
4. There were a lot of typos and grammar errors that could be amended carefully.
Author Response
Response to Reviewer 2 Comments
Point 1: Please use an unstructured form for the abstract per journal requirements. Please remove unnecessary abbreviations if they were not used more than three times. Line 29: Did you mean an "MCT" rather than an SRT?
Response 1: Thank you very much for your positive and professional comments on the manuscript. Unnecessary abbreviations were deleted. The abstract was brought to the proper format. The sentence in Line 29 was corrected
Point 2: The introduction was extraordinarily long. Please make it more concise. May consider providing a table that summarizes the difference between iCBT-I and CBT-I, and MCT and SRT. Otherwise, please use the same abbreviations throughout the manuscript.
Response 2: Thank you for this comment. The text was checked termwise and all the terms were streamlined. Unnecessary abbreviations were removed. The introduction was shortened where it was possible
Point 3: Please provide detailed information on the statistical software (R software?).
Response 3: The sentence about statistical software was added. Thanks again for taking your time to review our manuscript.
Reviewer 3 Report
The authors performed a secondary analysis of a randomized controlled trial comparing an online multicomponent CBT-I (MCT) and online sleep restriction therapy (SRT) for chronic insomnia patients. The found that high level of dysfunctional beliefs and attitudes about sleep (DBAS) was a moderator for better effects in the MCT, suggesting DBAS scale may be recommended to select patients for MCT rather than a SRT. Also, they identified certain factors had prognostic value for a better outcome in both MCT and SRT. The study is well-designed. I have several suggestions for the authors.
1. Line 133. The meaning of 4 "high education" in not specific. Based on the data presented in Table 1, it means university education. It is suggested to use the same term for both the main text and Table 1.
2. The legends of Table 2 and Table 3 are almost the same, and it is difficult to know the differences between the two tables. It is suggested to rephrase the table legends to make readers more clear about the differences of the two tables.
3. Some abbreviations seem not to be necessary as they appear only one time in the manuscript. For example, insomnia disorder (ID) and polysomnography (PSG).
Author Response
Response to Reviewer 3 Comments
Point 1: Line 133. The meaning of 4 "high education" in not specific. Based on the data presented in Table 1, it means university education. It is suggested to use the same term for both the main text and Table 1.
Response 1: Thank you for your positive decision on our paper and the helpful suggestions. We have revised the manuscript according to your suggestions to improve its quality. The term was specified and now you may see “university education” in both the main text and Table 1.
Point 2: The legends of Table 2 and Table 3 are almost the same, and it is difficult to know the differences between the two tables. It is suggested to rephrase the table legends to make readers more clear about the differences of the two tables.
Response 2: Table 2 name was changed to “Prognostic regression analysis coefficients”. Table 3 name – to “Predictive regression analysis coefficients”.
Point 3: Some abbreviations seem not to be necessary as they appear only one time in the manuscript. For example, insomnia disorder (ID) and polysomnography (PSG).
Response 3: These abbreviations were removed. Thanks again for taking your time to review our manuscript.